# Regulation of Pol II Pausing during Daily Gene Transcription in Mouse Liver

**DOI:** 10.3390/biology12081107

**Published:** 2023-08-09

**Authors:** Wei Xu, Xiaodong Li

**Affiliations:** College of Life Sciences, Wuhan University, Wuhan 430072, China; xw1049831018@126.com

**Keywords:** circadian rhythms, transcription, pausing, mammalian, mouse

## Abstract

**Simple Summary:**

Clock proteins and their collaborating transcription factors often act as distal enhancers to regulate the rhythmic transcription of gene promoters. Those transcription factors need to interact with the mediator complex and general transcription factors near the transcription start site to finally control transcription. Pol II pausing, which is determined by Pol II recruitment, pause release, and premature transcription termination near the transcription start site, plays a critical role in influencing the final transcription output. However, the way Pol II pausing is regulated during daily transcription and its role in shaping transcription rhythms have not been systematically investigated. We recently carried out a quantitative ChIP-seq study to characterize Pol II pausing across the day in mouse liver. Our analyses suggest that Pol II recruitment, pause release, and premature transcription termination activities exhibit genome-wide changes that can peak at distinct clock phases. Such complexity of Pol II pausing regulation during daily transcription provides new perspectives on the transcription regulation of circadian rhythms and warrants future studies to dissect the regulatory mechanisms of Pol II pausing and their roles in shaping the daily rhythms of gene transcription.

**Abstract:**

Cell autonomous circadian oscillation is present in central and various peripheral tissues. The intrinsic tissue clock and various extrinsic cues drive gene expression rhythms. Transcription regulation is thought to be the main driving force for gene rhythms. However, how transcription rhythms arise remains to be fully characterized due to the fact that transcription is regulated at multiple steps. In particular, Pol II recruitment, pause release, and premature transcription termination are critical regulatory steps that determine the status of Pol II pausing and transcription output near the transcription start site (TSS) of the promoter. Recently, we showed that Pol II pausing exhibits genome-wide changes during daily transcription in mouse liver. In this article, we review historical as well as recent findings on the regulation of transcription rhythms by the circadian clock and other transcription factors, and the potential limitations of those results in explaining rhythmic transcription at the TSS. We then discuss our results on the genome-wide characteristics of daily changes in Pol II pausing, the possible regulatory mechanisms involved, and their relevance to future research on circadian transcription regulation.

## 1. Introduction

Molecular clockworks consisting of feedback loops of core clock genes drive cell-autonomous circadian oscillation in various species [1]. In mammals, the transcription factors (TFs) CLOCK and BMAL1 dimerize to activate the transcription of *Per1/2* and *Cry1/2*, whose protein products are repressors that inhibit CLOCK/BMAL1 action through negative feedback [2]. While the post-translational regulation of clock proteins play critical roles in setting the clock pace [3,4], the prime mover of circadian oscillation is thought to be transcription [5]. High throughput technologies such as microarray [6], RNA-seq [7,8], and ChIP-seq [9] enable the detailed characterization of gene rhythms and the genomic binding of clock proteins, allowing for in depth analyses of circadian rhythm generation at the level of transcription. 

The binding sites of clock proteins are located within open chromatin regions established by tissue-specific pioneer TFs (tsTFs), and thus are typically tissue-specific [10]. Chromatin is known to be a barrier to transcription, and DNA sequences are often not accessible to many TFs, with the exception of tsTFs that are sufficient to trigger enhanced competency within chromatin. Furthermore, tsTFs allow subsequent binding by other TFs, including clock proteins. Some tsTFs (e.g., HNF4a [11,12]) and ubiquitous TFs (u-TFs, e.g., RELA/p65 [13,14]) interact with and recruit clock proteins to their cis elements. CLOCK/BMAL1 can also facilitate the binding of some tsTFs, leading to the suggestion that CLOCK/BMAL1 acts like a pioneer-like TF [10,15]. Like many TFs [16], clock proteins recruit cofactors to modify histones and remodel nucleosomes to regulate transcription. Clock proteins and their cofactors form a complex with an M.W. over 1 MDa, and deficiencies in some cofactors alter clock dynamics [17]. For example, clock proteins in both Drosophila and mammals recruit the TIP60 complex to regulate clock oscillation [18,19,20]. To control Pol II transcription at the transcription start site (TSS), TFs require the mediator complex [21] to interact with general transcription factors (GTFs) that are present at gene promoter [22]. The mediator subunits interacting with the clock protein complex remain to be determined. 

Traditional studies addressed how TFs and cofactors direct the mediator complex to assemble the pre-initiation complex (PIC) for transcription initiation and reinitiation at the TSS [23]. Distal enhancers bound by TFs and promoters are thought to be brought to proximity via chromatin looping, which is a process assisted by proteins such as cohesin and CTCF [24]. The traditional view of transcription, however, has difficulties in explaining new findings such as transcription bursting, which represents Pol II initiation and multiple rounds of reinitiation [25]. Imaging studies in single cells revealed that the transcription of many genes, including clock genes [26], is stochastic and of low frequency. Transcription often toggles between active and inactive states within a cell, and the active state is characterized by transcription bursting followed by a prolonged dormancy of the inactive state [27]. Bursting could be readily explained by the formation of a transcription hub: cluster and/or molecular condensate of TFs, cofactors, mediators, and Pol IIs that permit multiple rounds of Pol II initiation [28,29]. Recent studies revealed that TFs often contain intrinsically disordered low-complexity domains, whose interactions induce the formation of transcription hubs and even molecular condensates, the latter via “lipid-lipid phase separation” [28,29]. Transcription bursting also requires pause release [30], which refers to the process of P-TEFb-licensed Pol II elongation to overcome the +1 nucleosome barrier and transcribe into the gene body [31]. Initiated Pol II travels only a short distance; it then enters the state of pausing, wherein Pol II stays paused downstream of the TSS via the actions of pausing factors (DSIF and NELF) and the +1 nucleosome [31,32]. P-TEFb is a component of the super elongation complex (SEC) [33,34] that releases paused Pol II for elongation, permitting Pol II reinitiation to achieve transcription bursting. Initiated Pol II is also subject to premature transcription termination at the 5′ end of genes, which can decrease Pol II pausing [35,36].

Compared to other aspects of the transcription regulation of circadian rhythms [37], Pol II recruitment and pausing have just begun to attract attention from the circadian rhythm field [2,19]. Recently, we performed quantitative ChIP-seq analyses of Pol II recruitment and pausing during daily transcription in mouse liver, and revealed unique characteristics of those regulatory steps [38]. In this article, we first review the critical roles of rhythmic transcription in the generation of mRNA rhythms and clock oscillation, as revealed by historical and recent studies on transcription regulation by clock proteins and other TFs. We then discuss our findings on the daily regulation of Pol II pausing, which reveal the global characteristics of Pol II recruitment, pause release, and premature transcription termination at the 5′ end of genes. Given the critical role of Pol II pausing regulation in determining the transcription output, elucidating the mechanisms of Pol II pausing regulation during daily transcription is critical to understand the regulatory logic of circadian rhythm generation. 

## 2. Transcription Regulation Is the Main Driving Force for Gene Expression Rhythms

First demonstrated for *Per* in flies [39], core clock genes exhibit robust daily changes in their mRNA expression. Owing to rapid co-transcriptional splicing, the pre-mRNA level can be used as the surrogate for transcription activity. RNAse protection assays against *Per* pre-mRNA and mRNA showed that the *Per* mRNA rhythm in *Drosophila* is mainly driven at the level of transcription [40]. The post-transcriptional regulation of mRNA stability also contributes to the *Per* mRNA rhythm and is sufficient to confer rhythmic mRNA expression to other genes [41,42]. In mammals, core clock genes such as *Per1/2* also exhibit robust daily changes in mRNA levels [43,44]. A pre-mRNA measurement implicated that rhythmic transcription is the driving force for the mRNA rhythms of core clock genes and many other genes [45,46]. Deep sequencing studies evaluated the contribution of transcription regulation to mRNA rhythm generation in a genome-wide manner. One study estimated that 22% of mRNA rhythms are driven by rhythmic transcription [9]. Later studies with a high sequencing depth and kinetic modeling increased the estimate to about 70–80%, whereas rhythmic degradation contributes to the mRNA rhythms of 30–35% genes [47,48]. Nuclear export, which is another post-transcriptional regulation step, contributes to rhythm generation for 10% of rhythmic transcriptomes [49]. Overall, rhythmic transcription is deemed as the main driving force for gene rhythms [5].

## 3. Both the Intrinsic Tissue Clock and Extrinsic Cues Can Regulate Gene Expression Rhythms

Clock genes typically harbor multiple cis elements for clock proteins, which also have numerous other binding sites across the genome. Clock proteins thus also regulate many other genes. Clock genes and other genes are also influenced by extrinsic cues, which are often rhythmic in wildtype animals. Such cues include body temperature (T_b_) [50], feeding [51], and communication signals from other tissues (including the autonomic nervous system) [52]. The extrinsic cues can engage TFs as well as post-transcriptional mechanisms to regulate gene rhythms, including those of clock genes. For example, daily changes in T_b_ drive rhythmic HSF1 expression to regulate gene transcription [53]. The T_b_ rhythm also drives Cirbp expression to post-transcriptionally regulate clock dynamics [54]. Besides the T_b_ rhythm, blood-borne cues also regulate clock dynamics; serum and plasma can activate multiple signaling pathways to impact clock genes [55,56,57]. For example, rhythmic cues in plasma activate SRF, which regulates the transcription of the clock gene *mPer2* [57]. Certain blood-borne cues impacting clock dynamics are heat labile, implying that they are proteins [58]. Lipids can also serve as inter-tissue communicating cues. For example, phosphatidylcholine is synthesized by the liver and released into plasma to activate PPARα in muscles [59]. 

Overall, clock proteins and many other TFs exhibit daily changes in their actions. Like clock proteins, other TFs also have thousands of genomic binding sites in various tissues. Therefore, they potentially can regulate numerous genes besides clock genes. Gene expression rhythms are thus driven by both clock proteins and other TFs. How clock proteins and other TFs work together to control gene rhythms was the focus of recent studies in various peripheral mouse tissues.

## 4. Clock Proteins Typically Collaborate with Other TFs to Regulate Transcription Rhythms

Clock proteins and other TFs often collaborate to regulate target genes [10]. The independent contribution of the clock to gene rhythms is rather limited [60,61]. In studies that reconstitute clock oscillation (RE) in specific tissues of *Bmal1*-deficient mice [60,61], it was shown that only 10% of the rhythmic transcriptome can be restored in the livers of liver-RE mice [60]. However, that is not to say that the liver clock regulates only 10% of the rhythmic transcriptome in wildtype mice. In fact, the disruption of the liver clock disturbs about 90% of the gene rhythms in mouse liver [62,63]. Overall, those results indicate that the majority of gene rhythms are regulated in a combinatorial manner by both the intrinsic clock and the TFs engaged by extrinsic cues. 

By comparing the liver gene rhythms in *Bmal1* KO, liver-RE, and wildtype mice under ad libitum feeding versus nighttime restricted feeding, it was shown that the mRNA rhythms in the livers of the wildtype mice can be partitioned into four parts based on their modes of regulation [64]. Some rhythms can be driven by the intrinsic liver clock alone (13.7%); some can be driven by rhythmic feeding cues alone (17.5%); some require not only the intrinsic clock, but also rhythmic feeding cues (34.5%); while the rest (34.4%) require both the intrinsic clock and rhythmic cues from other tissues (and their clocks). Those results indicate that for the regulation of a majority of gene rhythms, there is a mandatory requirement for clock proteins to collaborate with other TFs. For example, feeding engages the TF CEBPB to coregulate BMAL1 target genes, and CEBPB deficiency disrupts the rhythms of some BMAL1 target genes that are also regulated by feeding [64]. 

## 5. The Need to Study Pol II Pausing Regulation near the TSS

Clock proteins and other TFs occupying distinct enhancers of the same gene can collaborate through chromatin looping to regulate transcription. Techniques such as Hi-C and CHIA-PET revealed daily changes in the long-range interactions between distinct enhancers bound by clock proteins and other TFs, respectively, and between those enhancers and gene promoters [10,65]. The collaboration between clock proteins and other TFs can also occur at the same enhancers. Indeed, TFs often exhibit cooperative binding at the same enhancers to increase the affinities of the two factors to their respective motifs. However, cooperative binding does not necessarily lead to coactivation. For example, HNF4a and RELA/p65 can recruit CLOCK/BMAL1 for genomic binding [11,13], but can transrepress its transcription activation [12,14]. Such interactions between TFs at same and/or distinct enhancers pose serious challenges in elucidating how clock proteins contribute to the final transcription output at the TSS. Indeed, the genomic binding of CLOCK/BMAL1 at enhancers is often not sufficient to specify the rhythm phase and amplitude, and cannot confer rhythmicity to some target genes [37].

Another aspect of the complexity of transcription regulation is the lack of consensus on how TFs and their cofactors at distal enhancers regulate transcription near the TSS [66]. The textbook model of chromatin looping posits that a stable contact is formed between distal enhancers and promoters. A variation of this classical model is the “kiss-and-run” model of transient contact between distal enhancers and promoters. However, the nature of such long-range genomic interactions and its relevance to transcription have recently been questioned [66]. The alternative TAG (TF activity gradient) model [66] emphasizes contact-independent “communication by diffusion” of TFs and their cofactors between enhancers and promoters. However, given the diversity of interacting TFs and their multitudes of cofactors, it would be difficult, if not impossible, to dissect the specific contribution of an individual TF and/or cofactor to the transcription output. On the other hand, Pol II recruitment and Pol II pausing represent the final regulatory outcomes by a plethora of TFs and cofactors. Those regulatory steps are directly related to the final transcription output. Obtaining information about them is thus critical for understanding the logic of transcription regulation. Surprisingly, such information is lacking in circadian rhythm research (Figure 1). 

Against this backdrop, we performed a ChIP-seq study of the Tbp (TATA-binding protein. A TFII D subunit) and Pol II during daily transcription in mouse liver [38]. We used the Tbp to measure the Pol II recruitment at the gene promoter and assumed that the Tbp signal near the TSS is proportional to the rate of Pol II initiation (and reinitiation). However, the Tbp and the mediator remain promoter-bound during PIC formation and Pol II initiation and reinitiation [67,68], while other GTFs such as TFII B dissociate after Pol II initiation and recycle for Pol II binding during reinitiation [69,70]. Thus, relative to the signals of other GTFs, the Tbp signal might overestimate the Pol II initiation and reinitiation rates. Nonetheless, the Tbp and other GTFs appear to exhibit concordant changes near the TSS [71,72], permitting our use of the Tbp signal to measure not only Pol II recruitment, but also initiation and reinitiation [73]. The Tbp signals within the TSS region (defined as −50 to +300 bp to TSS [38,74]) were quantitated to measure Pol II recruitment ([Tbp]_TSS_). The Pol II ([Pol II]_TSS_) signals within the TSS region were quantitated to measure the paused Pol II, while the Pol II signals in the gene body ([Pol II]_GB_) were quantitated to measure the gene transcription rate. The Pol II traveling ratios (TR: [Pol II]_TSS_:[Pol II]_GB_), which are quantitative measures of Pol II pausing [74,75], were also calculated. By means of the systematic characterization of Pol II recruitment and pausing for 7414 genes during daily transcription, our study provides the first glimpse of their genome-wide characteristics. 

## 6. Global Characteristics of Pol II Recruitment and Pausing during Daily Transcription in Mouse Liver

The results of our study are summarized in Figure 2. As can be seen, Pol II recruitment measured via [Tbp]_TSS_ is typically low during nighttime, especially at ZT22 (zeitgeber time 22. ZT0 corresponds to the lights that are on during the 12:12 light/dark cycle). [Tbp]_TSS_ exhibits a great rebound at ZT2. Most genes also have higher [Pol II]_TSS_ near ZT2. Moreover, numerous genes’ transcription rates, as measured via [Pol II]_GB_, are higher at ZT2, which is concurrent with the increase in [Pol II]_TSS_ and [Tbp]_TSS_ (Figure 2). Together, those results indicate a global upregulation of gene transcription at ZT2. Nonetheless, [Pol II]_GB_ exhibits more gene-specific changes across the day than [Pol II]_TSS_ and [Tbp]_TSS_, with many genes’ transcription peaking at other time points than ZT2, especially near ZT14. The overall bimodal distribution of peak gene transcription is consistent with the pre-mRNA analysis results obtained by others [48]. 

The characteristics of daily changes in [Tbp]_TSS_, [Pol II]_TSS_, and [Pol II]_GB_ (Figure 2) may be related to the cell cycle, whose progression is known to impact transcription [76]. In particular, transcription is generally inhibited during mitosis and reactivated upon mitotic exit [77,78]. While the liver is typically considered a non-dividing organ, it still exhibits daily changes in cell-cycle related activities that interplay with the clock [79,80]. For example, the activity of CDK1, which is critical for mitosis entry, exhibits daily changes controlled by the clock [81] and also regulates clock oscillation in return [82]. CDK1 activity peaks before ZT0 in mouse liver [82,83]. CDK1 is known to phosphorylate TFII D to inhibit PIC formation [84]. The late-night rise in CDK1 activity in mouse liver could induce the global diminishment of [Tbp]_TSS_, as indeed observed at ZT22. On the other hand, the rebound of [Tbp]_TSS_ and [Pol II]_TSS_ at ZT2 could be analogous to gene reactivation upon mitotic exit [77,78]. 

Our results show that the Pol II TRs of all genes exhibit daily changes (Figure 2). The TRs of most genes are high near ZT0, and their nadirs are near ZT12, especially at ZT14 (Figure 2). [Pol II]_TSS_ also reaches its genome-wide nadir at ZT14 (Figure 2). Pause release lowers the Pol II TR by decreasing [Pol II]_TSS_ and increasing [Pol II]_GB_. The patterns of daily changes in [Pol II]_TSS_ and TR in our results suggest a global rhythm of pause release affecting most liver genes. This possibility is supported by other evidence. During pause release, P-TEFb recruits the PAF1 complex (PAF1c) [85] to stimulate the activity of SET1 [86,87], which deposits H3K4me3 downstream of the TSSs of genes. H3K4me3 is increased in a genome-wide manner in mouse liver in the early night [9,88,89], which is consistent with the rise in the pause release activity at that time. Against this global trend, however, some genes’ TRs peak near ZT14. The possible causes for such exceptions are discussed in Section 8.

The results of our ChIP-seq study are population averages lacking single-cell resolution. In single cells, the inherent noises of stochastic gene expression lead to phenotypic variations, such as period heterogeneity among clonal cell populations [90,91,92]. More specifically to our study, Pol II recruitment (and subsequent pausing) and pause release could be distinct events occurring randomly among cells. However, single cell imaging studies showed that Pol II recruitment and bursting are not mutually independent, but are sequential events occurring in close succession [93]. This permits cross analyses of the Tbp and Pol II signals to infer the rules of daily transcription regulation. Below, we discuss our analysis results and their implications.

## 7. Pol II Recruitment Is Not a Direct Determinant of Gene Transcription Rate

Traditionally, Pol II recruitment is thought to be the determinant of transcription output. However, our results show that [Tbp]_TSS_ does not correlate well with [Pol II]_GB_ for numerous genes [38]. Six example genes are shown in Figure 3. While the [Tbp]_TSS_ values of all six genes are the highest at ZT2, the transcription rates ([Pol II]_GB_) of those genes peak at different phases. For example, the [Tbp]_TSS_ and [Pol II]_GB_ values of *Bmal1* are the highest at ZT2, but several genes’ transcription rates peak near ZT14, when their [Tbp]_TSS_ values are low compared to other time points. Such results clearly indicate that Pol II recruitment does not directly determine the transcription rate. This appears to be at odds with the current view of coordinated regulation of Pol II recruitment and transcription bursting [93]. Nonetheless, the paradox can be reconciled if pause release, which is required for transcription bursting, is regulated independently from PIC formation. Such a scenario was reported in [94]. Upon the acute depletion of the mediator complex to limit Pol II recruitment and initiation, the cell-type-specific genes’ transcription is lowered [94]. However, the transcription of many other genes is maintained due to a compensatory rise in pause release [94]. Such results indicate that pause release can affect transcription in a manner independent of Pol II recruitment.

Transcription bursting has two parameters, burst frequency and burst size, which often respond differentially to biological stimuli and experimental manipulations [95]. Burst frequency is primarily determined via Pol II recruitment, which leads to PIC formation and transcription initiation [96]. On the other hand, the burst size is mainly affected by pause release, which permits rounds of reinitiation. If pause release is low during Pol II recruitment, then Pol II from the first initiation round would stay paused downstream of the TSS, creating a barrier for trailing Pol II. Eventually, it can hinder the downstream movement of Pol II in the PIC. Indeed, Pol II pausing inhibits initiation [71,72]. By contrast, the P-TEFb-mediated release of paused Poll promotes Pol II reinitiation to increase the burst size. It might also reduce the dwelling time of the Pol II “hub” near the TSS to facilitate new PIC formation and initiation. Thus, pause release might increase burst frequency. With regard to daily transcription in mouse liver, we suspect that genes with the highest [Tbp]_TSS_, [Pol II]_TSS_, and [Pol II]_GB_ at ZT2 have high burst frequencies but low burst sizes at this time point. By contrast, the upregulation of some genes’ transcription near ZT14 may be due to increased burst sizesassociated with high pause release activity. Such a scenario is supported by other evidence, such as the genome-wide increase in H3K4me3 in mouse liver in the early night [9,88,89].

TFs, their cofactors, and the mediator complex are known to regulate pause release through interactions with P-TEFb. TFs such as c-Myc [75] can directly recruit P-TEFb. The cofactor BRD4 also binds P-TEFb [97,98], and the TIP60 complex is suggested to acetylate BMAL1 to recruit BRD4-P-TEFb [19]. The mediator complex not only regulates PIC formation, but also regulates downstream transcription events, including Pol II initiation and pause release [25]. The MED23 and MED26 subunits and the CDK8 mediator kinase module (MKM) of the mediator complex have been shown to interact with P-TEFb/SEC [99,100]. However, we want to emphasize that, in the molecular interactions controlling pause release, P-TEFb is probably a rate-limiting factor. P-TEFb can be sequestered into the 7SK snRNP, where it remains inactive and unable to elicit pause release. Signaling pathways can activate pause release via inducing 7SK snRNP disassembly to release P-TEFb [94,101,102]. As shown in [94], the disassembly of 7SK snRNP boosts the P-TEFb availability to increase pause release. This can be a potential mechanism accounting for daily changes in pause release activity in mouse liver.

## 8. Premature Transcription Termination at the 5′ End of Genes Contribute to the Regulation of Pol II Pausing

Pause release decreases Pol II stability in the TSS region by enabling Pol II outflux for elongation. Footprinting and imaging studies revealed a rapid turnover of Pol II near the TSS [103,104]. However, blocking pause release via P-TEFb inhibition only partially increases Pol II stability [103,104,105]. Those results implicate that transcription termination at the 5′ end of genes is the major determinant of Pol II stability near the TSS [35,36]. Such premature termination is mediated by factors such as XRN2 [106], which also terminates Pol II transcription at the 3′ end of genes [107]. Interestingly, clock proteins can inhibit Pol II termination at the 3′ ends of the clock genes *Per1* and *Cry2*, leading to daily changes in Pol II accumulation therein [108]. To characterize whether premature termination at the 5′ end of genes changes over the day and to determine its contribution to Pol II stability near the TSS, we used the ratio of [Pol II]_TSS_ to [Tbp]_TSS_ as the index of Pol II stability within the TSS region. As evident in Figure 2, Pol II stability is the lowest at ZT2, when the pause release activity appears to be low. On the other hand, Pol II stability at ZT14, which is a time point that presumably has high pause release activity, is intermediate among the six daily time points (Figure 2). Those results indicate a critical role of premature termination in lowering Pol II stability [35,36]. Because premature termination decreases [Pol II]_TSS_ (thus Pol II TR) in a manner independent of P-TEFb, it leads to an inaccurate estimate of the pause release. For example, the Pol II TRs of most genes are low at ZT14, which is most probably due to a global increase in the pause release. However, some genes’ TRs peak at ZT14 (Figure 2). We suspect that, for those outlier genes, premature termination might significantly lower their TR values at other time points to confound the estimate of pause release activity at ZT14.

The TSS region can only harbor a limited number of Pol IIs, and such space limitation is suggested to subject Pol IIs to collisions that promote premature termination [109]. However, the target of premature termination needs clarification. In ChIP-seq studies with a high sequencing depth, two Pol II peaks can be observed in the TSS region [110]. The first peak centers on the TSS and represents the initiated Pol II before pausing. The second peak is 110 bp downstream of the TSS and represents canonical pausing. Existing evidence suggests that the paused Pol II (the second Pol II peak) is stable. For example, following triptolide treatment to inhibit Pol II initiation, the half-lives of Pol IIs near the TSS are typically minutes to even above an hour [111,112], indicating that paused Pol IIs are not prone to rapid turnover via premature termination. While such long half-lives were questioned based on the efficacy of triptolide treatment [103,104], we suspect that premature termination mainly targets initiated Pol II before pausing. The capping of nascent pre-mRNAs starts upon transcription initiation and is completed when Pol II enters the pausing state [113]. Nascent RNAs with a 5′ cap are very stable, with only about 1% being subjected to premature termination, indicating that Pol II pausing is stable [114]. 

The pausing factor NELF recruits the cap-binding complex (CBC) to bind the m7G cap [115]. Importantly, the m7G cap and its binding by CBC are checkpoints for pre-mRNA splicing and Pol II elongation [116,117,118]. This suggests a quality control role of premature termination to ensure productive Pol II elongation. Indeed, premature termination involves pre-mRNA quality control mechanisms. For example, XRN2 plays a role in premature termination [106]. XRN2 acts on uncapped RNAs, and its action is assisted by decapping enzymes such as DXO and DCP2, whose likely targets are inappropriately capped pre-mRNAs [119]. The integrator complex (INTS), which cleaves nascent RNAs and recruits PP2A to dephosphorylate Pol II CTD, also functions in premature termination [120]. INTS depletion leads to the production of unspliced transcripts by Pol IIs that are incompetent for productive elongation [121,122], indicating that INTS functions in quality control to ensure productive Pol II elongation and efficient co-transcriptional pre-mRNA splicing. By contrast, CBC functions in P-TEFb recruitment and INTS exclusion to activate pause release and productive elongation [123,124]. 

## 9. Conclusions

To regulate gene transcription rhythms, clock proteins and their collaborating TFs at distal enhancers need to gain access to the mediator complex and GTFs near the TSS. Pol II recruitment, pause release, and premature transcription termination are the three processes that control Pol II pausing and the transcription output near the TSS. Our previous study revealed that those three processes exhibit genome-wide changes that could peak at distinct clock phases, thus providing new perspectives on the logic of the transcription regulation of circadian rhythms. Future studies should be directed towards elucidating the mechanisms for daily changes in pause release and premature termination and towards pinpointing their roles in shaping the daily rhythms of gene transcription. A major limitation of our study and many others is the routine use of [Pol II]_TSS_ as the measure of paused Pol II. However, [Pol II]_TSS_ actually contains signals from multiple Pol II forms [110], including PIC, initiated Pol II before pausing, and paused Pol II. [Pol II]_TSS_ thus overestimates paused Pol II. Moreover, premature termination can lower [Pol II]_TSS_ (and TR) independent of pause release. Those confounding factors can lead to inaccuracies in the TR results. Ideally, only signals of paused Pol II should be used for TR calculation to measure Pol II pausing and pause release. However, it is not feasible in practice to find an antibody that exclusively recognizes paused Pol II. Even monoclonal antibodies against a specific Pol II CTD form (e.g., pSer5) show various cross-reactivities to other forms, and different antibodies against the same epitope can yield dramatically different results [125]. We suggest alternative targets such as CBC components [115] as surrogates of paused Pol II to improve accuracies in the quantitative analyses of Pol II pausing and pause release in future studies.

## Figures and Tables

**Figure 1 biology-12-01107-f001:**
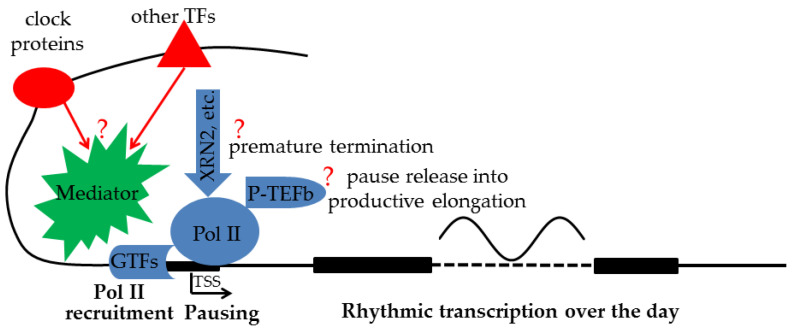
While clock proteins collaborate with other TFs at distal enhancers to regulate rhythmic transcription of target genes, exactly how final transcription output is determined by Pol II recruitment, premature termination, and pause release activities near the TSS is still an open question (?) and needs to be systematically characterized.

**Figure 2 biology-12-01107-f002:**
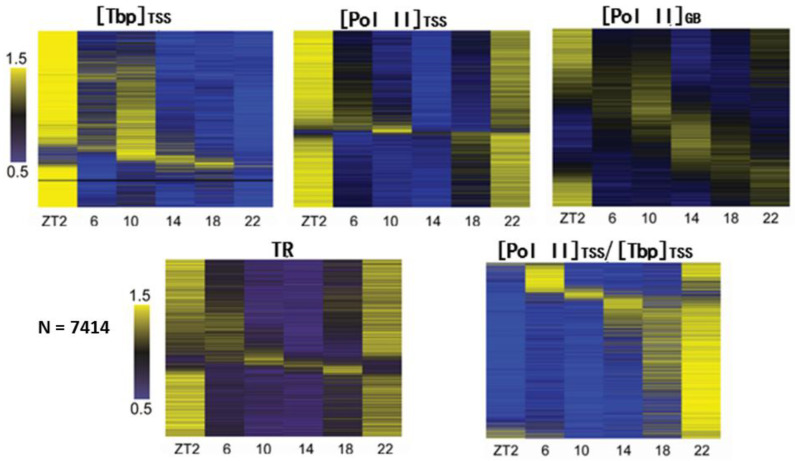
Summary of our results on daily regulation of Pol II recruitment and pausing near the TSS of mouse liver genome. In total, 7414 genes were finally chosen for analyses. The top panel shows results of [Tbp]_TSS_, [Pol II]_TSS_, and [Pol II]_GB_ in heatmaps. The bottom panel shows daily changes of Pol II TR and stability within the TSS region ([Pol II]_TSS_/[Tbp]_TSS_).

**Figure 3 biology-12-01107-f003:**
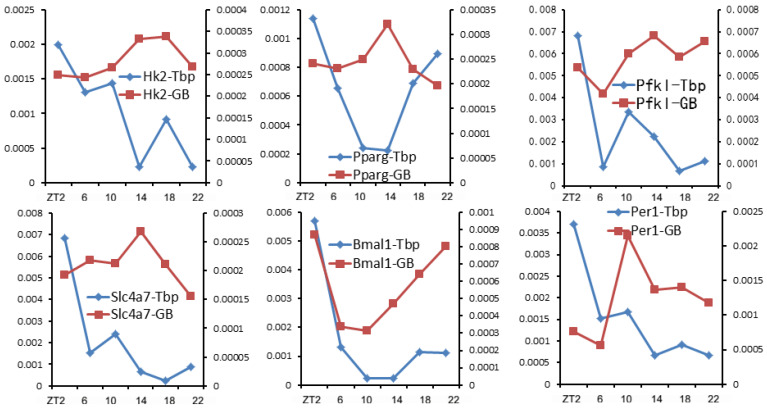
Pol II recruitment ([Tbp]_TSS_; left *y*-axis) and gene transcription rate ([Pol II]_GB_; right *y*-axis) of example genes. It is evident that Pol II recruitment is not always a direct determinant of transcription rate.

## Data Availability

The data presented in this manuscript script are based on GSE96773 in Gene Expression Omnibus.

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
