# Peer review of "Regulation of Pol II Pausing during Daily Gene Transcription in Mouse Liver"

_biology, 2023, doi:10.3390/biology12081107_

Round 1
Reviewer 1 Report
The review by Xu et al describes extensively the distinct transcription-related mechanisms contributing to generation of circadian rhythms in eukaryotes to introduce their own recent results concerning the measure of RNA Pol II pausing along 24h and results showing its involvement in daily transcription.
The review is quite synthetic, well written and interesting for readers from the circadian field.
My only concern : it would be useful that the theoretical model based on data regarding PolII pausing during the 24h cycle in the liverand that is described in the review, also is illustrated by a schematic figure.
The quality of the text is very good; a few typing errors should be corrected.
Author Response
Response: In the revised manuscript, we corrected all errors in writing and added a new figure as suggested (also by another reviewer). Per the editor’s request, we also added a “Simple Summary” section and a “Conclusions” section. As a result of manuscipt restructuring, we rewrote section 5, 7&9 during revision.
Reviewer 2 Report
Xu and Li propose a literature review covering circadian rhythms, Pol-II elongation pausing, and their results with the mouse liver. The language is accessible and several sections are interesting and informative. Unfortunately, the review does not have clear aims, which leads to redundancies and other topics quite overlooked. An example is the differences between the title, the aims in the abstract (lines 14-17), and the aims at the end of the introduction (lines 74-78).
Another concern is that figures only cover the data from the authors. There are many complex mechanisms described here that could be presented in figures (e.g. elongation pausing, control of circadian genes)
Author Response
Xu and Li propose a literature review covering circadian rhythms, Pol-II elongation pausing, and their results with the mouse liver. The language is accessible and several sections are interesting and informative. Unfortunately, the review does not have clear aims, which leads to redundancies and other topics quite overlooked. An example is the differences between the title, the aims in the abstract (lines 14-17), and the aims at the end of the introduction (lines 74-78).
Response:. In the revision, we rewrote the abstract and introduction to make our aims more specific by focusing on the relevance of Pol II pausing regulation near the TSS to the generation of transcription rhythms. In section 8 of the revised manuscript, we pointed out that transcription termination at the 3’-end of genes is also critical for clock oscillation, although we did not have an in-depth discussion on this topic (and others).
Another concern is that figures only cover the data from the authors. There are many complex mechanisms described here that could be presented in figures (e.g. elongation pausing, control of circadian genes).
Response: We added a new figure as suggested (also by another reviewer). Per the editor’s request, we also added a “Simple Summary” section and a “Conclusions” section. As a result of manuscipt restructuring, we rewrote section 5, 7&9 during revision.
Round 2
Reviewer 2 Report
Xu and Li addressed partially the concerns made during the first round of review. Unfortunately, most concerns remain and the manuscript does not seem eligible for publication in the current version.